# Symbiotic Functioning and Photosynthetic Rates Induced by Rhizobia Associated with Jack Bean (*Canavalia ensiformis* L.) Nodulation in Eswatini

**DOI:** 10.3390/microorganisms11112786

**Published:** 2023-11-16

**Authors:** Zanele D. Ngwenya, Felix D. Dakora

**Affiliations:** 1Department of Crop Sciences, Tshwane University of Technology, Private Bag X680, Pretoria 0001, South Africa; ngwenyazd@tut.ac.za; 2Department of Chemistry, Tshwane University of Technology, Private Bag X680, Pretoria 0001, South Africa

**Keywords:** root nodulation, nodule number, nodule dry mass, ^15^N/^14^N isotopic analysis, N-fixed, % relative symbiotic effectiveness, gas-exchange, photosynthesis

## Abstract

Improving the efficiency of the legume–rhizobia symbiosis in African soils for increased grain yield would require the use of highly effective strains capable of nodulating a wide range of legume plants. This study assessed the photosynthetic functioning, N_2_ fixation, relative symbiotic effectiveness (%RSE) and C assimilation of 22 jack bean (*Canavalia ensiformis* L.) microsymbionts in Eswatini soils as a first step to identifying superior isolates for inoculant production. The results showed variable nodule number, nodule dry matter, shoot biomass and photosynthetic rates among the strains tested under glasshouse conditions. Both symbiotic parameters and C accumulation differed among the test isolates at the shoot, root and whole-plant levels. Although 7 of the 22 jack bean isolates showed much greater relative symbiotic efficiency than the commercial *Bradyrhizobium* strain XS21, only one isolate (TUTCEeS2) was statistically superior to the inoculant strain, which indicates its potential for use in inoculant formulation after field testing. Furthermore, the isolates that recorded high %RSE elicited greater amounts of fixed N.

## 1. Introduction

Soil nutrient depletion is the major biophysical cause of declining per capita food production in Sub-Saharan Africa [1,2,3]. Over the past few decades, small-scale farmers in Africa have lost large quantities of nutrients from soils due to depletion by crop species, especially with the lack of input of manure or fertilizers [3,4]. An increase in fertilizer use has therefore been recommended if Africa is to meet its food security needs and reduce poverty [5]. However, the high cost of chemical fertilizers is a major constraint to their use by resource-poor farmers [6,7]. As a result, smallholder farmers in Africa have resorted to the use of organic fertilizers such as farmyard manure, or relied on seasonal fallow and legume rotations for building soil fertility [8,9,10].

Legumes are a major source of biological nitrogen [10,11] and play a crucial role in maintaining soil productivity in smallholder agriculture in Africa [7]. Through the ability of bacteroids to reduce atmospheric N_2_ to NH_3_ in root nodules, legume plants are an option for use in sustainable agriculture in place of chemical N fertilizers [12]. About 70% of legume species are able to enter into a symbiotic relationship with compatible strains of N_2_-fixing rhizobia in soils [13,14]. The legume/rhizobia symbiosis can contribute about 80% of N to agricultural systems [15], as well as to soil health. However, this bacterial symbiosis with legumes is influenced by many factors including salinity, drought, nutrient imbalances and extreme temperatures [16,17]. Although high soil N can suppress nodulation [12,18,19], factors such as geographic location, host-plant genotype, soil type, as well as the rhizobial symbiont itself can also affect the efficiency of the symbiosis [16].

Bacterial inoculants made from highly effective rhizobia have been used to improve nodulation and N_2_ fixation in legumes, especially in fields without any history of legume cultivation [20,21]. According to [22], some rhizobial strains are capable of nodulating a limited number of host plants, while others are promiscuous and can nodulate a wide range of legumes. Agronomically, the use of highly effective rhizobia that nodulate many host plants is desirable [15,21].

Over the years, farmers have become more receptive to the use of inoculants due to the availability of high-quality products and multi-purpose elite rhizobia in the market place, as well as the improved grain yields obtained at a low cost when compared to the use of chemical N fertilizers [23]. In Sub-Saharan Africa, legume production has been constrained by a number of factors such as low soil fertility, soil acidity, soil salinity, and soil organic matter depletion [24]. Efforts have therefore been made to improve the efficiency of beneficial native microbes present in soils, as well as to identify adapted new elite strains for use as inoculants in stressful environments [23]. This has been achieved through the collection of root nodules, followed by bacterial isolation, and the characterization and assessment of isolate symbiotic efficiency in comparison with standard commercial inoculants [25].

Jack bean (*Canavalia ensiformis* L.) is one of the most underutilized food legumes [22]. Despite being neglected by most producers and researchers, jack bean seeds are a good source of dietary protein and carbohydrates; they contain approximately 600 g kg^−1^ carbohydrate and 300 g kg^−1^ protein [22,26]. Though, their young pods and beans are eaten as vegetables, they do contain anti-nutritional factors (protease inhibitors, lectins, saponins and tannins) that require careful processing [27]. Jack bean can also contribute about 360 kg N ha^−1^ to the cropping system through biological nitrogen fixation [28], and is grown as a green manure [26,29], cover crop or a companion to a cereal crop [27,30]. Numerous studies have reported on the potential of native African rhizobia-nodulating legumes such as Bambara groundnut, common bean, cowpea and Kersting’s groundnut, as well as introduced soybean, to be included in inoculant production for use across Africa through studying the symbiotic effectiveness of these legumes [15,31,32,33,34,35,36]. So far, however, no study has reported the symbiotic effectiveness of jack bean microsymbionts in Africa. The aim of this study was to identify highly effective rhizobial strains with potential for use in the development of inoculants for increased jack bean production. To achieve this, the symbiotic efficacy and capacity to induce photosynthetic function of diverse native rhizobial symbionts isolated from jack bean grown in contrasting agroecologies in Eswatini were assessed.

## 2. Materials and Methods

### 2.1. Origin of Isolates and Trapping Rhizobia in the Glasshouse

The rhizobia from root nodules of jack bean (*Canavalia ensiformis* L.) (Accesion 493) were trapped in a glasshouse. Soils from four locations (Malkerns Research Station (26°33′ S, 31°10′ E), Ka-Zulu (26°45′ S, 31°15′ E), New Heaven (27°03′ S, 31°29′ E) and Luve (27°03′ S, 31°29′ E)) in the different agroecological zones (Middleveld, Lower Middleveld and Highveld) of Eswatini were used to grow rhizobia-nodulating jack bean. The chemical properties of the soils used in this study, including available P, were measured as described by [37] and are presented in Table 1.

Three surface-sterilized [38] seeds were planted in autoclaved sand contained in 1200 cm^3^ pots and thinned out to one seedling per pot after germination. Three replicate pots were used per soil sample. Soil inocula were prepared by adding 20 g of each soil sample to 1000 mL sterile distilled water [22,33]. Thereafter, each pot was inoculated with soil suspension (about 5 mL) from the different locations at planting. The plants were irrigated with distilled sterile N-free nutrient solution [39], which was alternated with sterile water if necessary. At 60 days after planting (DAP), the plants were uprooted, and the nodules were removed, counted and stored in silica gel at 4 °C prior to bacterial isolation.

### 2.2. Bacteria Isolation from Root Nodules

The isolation of bacteria from root nodules of jack bean plants was performed as described by [40]. Healthy nodules were selected and the bacteria isolated according to [33]. The nodule suspensions were streaked onto yeast–mannitol agar (YMA) plates and incubated at 28 °C. The plates were monitored daily to record the time taken for colonies to appear. In instances where we did not get single colonies, re-streaking was done to obtain single colonies.

### 2.3. Authentication of Bacterial Isolates

These bacterial isolates were authenticated as described in [22]. Briefly, the jack bean isolates were examined for their ability to form root nodules on their homologous host in a naturally lit glasshouses with uncontrolled day/night temperatures. Prior to planting, seeds were surface-sterilized by immersing them in 95% ethanol for 5 to 10 s, then in 3% sodium hypochlorite for 2–3 min, and thereafter, the seeds were rinsed six times with sterile distilled water [40] and germinated in autoclaved sand contained in plastic pots (1200 cm^3^) covered with sterile non-absorbent cotton wool to avoid moisture loss and contamination by external bacteria. Three sterilized seeds were planted per pot and thinned to one seedling per pot after germination. The jack bean seedlings were inoculated with 1 mL of broth culture of the different bacterial isolates grown to the exponential phase using a sterilized micropipette. The plants were watered with N-free nutrient solution [39] and sterile water when necessary. Pots containing uninoculated plants, plants treated with 5 mM KNO_3_ and plants inoculated with the commercial *Bradyrhizobium* strain XS21were included as controls. Three replicate pots were used per isolate. The plants were harvested at 60 days after planting and assessed for nodulation.

### 2.4. Gas Exchange Measurements and Symbiotic Effectiveness

Photosynthetic rates (*A*), stomatal conductance (*gs*) and transpiration rates (*E*) of young and fully expanded trifoliate leaves of jack bean plants were measured using a portable infrared gas analyzer, version 6.2 (LI 6400XT, Lincoln, NE, USA), with the chamber conditions of photosynthetic flux density of 1000 µmol m^−2^ s^−1^, CO_2_ concentration of 400 µmol mol^−1^ and flow rate of 500 µmol s^−1^ to assess the photosynthetic effectiveness induced by the bacterial isolates. Three replicate leaves were measured per plant for each replicate pot (n = 9). An instant measure of water use efficiency (WUE) was computed via the ratio of *A* to *gs* [41]. Three replicate pots were used per isolate. The relative symbiotic effectiveness (RSE), defined as the strain symbiotic efficiency, was determined by expressing the shoot dry matter of plants inoculated with the test isolates as a percentage of the biomass produced by the plants inoculated with the commercial inoculant [42,43,44].
$$\%RSE=\frac{Shoot\;dry\;matter\;of\;inoculated\;test\;strain}{Shoot\;dry\;matter\;of\;inoculated\;standard\;strain}\times 100$$

### 2.5. Shoot and Root ^15^N/^14^N Analysis

Shoot and root ^15^N isotopic analyses were performed at the Stable Light Isotope Laboratory, University of Cape Town, South Africa. Approximately 2.5 mg samples of the ground shoot and root of inoculated and uninoculated jack bean plants were weighed into tin capsules and fed onto a Carlo Erba NA1500 Elemental Analyzer coupled to a Finnigan MAT 252 Mass Spectrometer (Finnigan MAT GmbH, Bremen, Germany) via a Conflo II Open-Split Device. An internal standard of *Nasturtium* spp. was included after every five runs of the plant samples in order to correct for machine errors associated with the isotopic analysis. The δ^15^N values of the jack bean plants were calculated [45]:(1)$$\delta {}^{15}N(‰)=\frac{{\left[{}^{15}N/{}^{14}N\right]}_{sample}-{\left[{}^{15}N/{}^{14}N\right]}_{atm}}{{\left[{}^{15}N/{}^{14}N\right]}_{atm}}\times 1000$$
where ^15^N/^14^N_sample_ is the abundance ratio of ^15^N and ^14^N in the sample, and ^15^N/^14^N_atm_ is the abundance ratio of ^15^N and ^14^N in the atmosphere.

The whole-plant δ^15^N was calculated as an average of the δ^15^N natural abundance values of all plant parts (shoot + roots) weighed according their respective total N contents [46]. The %N in the shoots and roots was obtained directly using a mass spectrometer, and the shoot and root N contents were calculated as the product of %N and shoot or root dry matter. 

### 2.6. Amount of N-Fixed in Shoots

The amount of N-fixed in shoots and roots was calculated as:N-fixed = N content of inoculated plant − N content of uninoculated control plants(2)

### 2.7. Shoot and Root ^13^C/^12^C Isotopic Analysis

To determine the ^13^C/^12^C ratios, the shoot and root samples were weighed into tin capsules and ran on the mass spectrometer, as described for the ^15^N/^14^N isotopic analysis. Pee Dee Belemite (PDB) was included as a universal standard for the limestone formulation. The ratio of ^13^C/^12^C in each sample was used to calculate the ^13^C natural abundance or δ^13^C (‰) as [47]:(3)δ13C=13C12Csample13C12Cstandard−1×1000
where ^13^C/^12^Csample is the isotopic ratio of the sample and [^13^C/^12^C]standard is the isotopic ratio of PDB, a universally accepted standard for limestone formation [48].

### 2.8. Statistical Analysis

The data collected were tested for normal distribution before being subjected to a one-way analysis of variance using Statistica (version 10.1). The Duncan’s Multiple Range test was used to separate means that showed significant differences at *p* ≤ 0.05. A correlation analysis was performed to assess the relationship between measured parameters.

## 3. Results

Few nodules were obtained from some of the soils used for trapping. For example, only one nodule was obtained from Ka-Zulu, while four nodules were obtained from New Heaven.

Root nodules obtained from trapping were used for bacterial isolation. A total of 45 isolates were obtained, 20 from Malkerns Research Station, 20 from Luve, 4 from New Heaven and 1 from Ka-Zulu. In the fulfilment of Koch’s postulate, 34 out of the 45 isolates could form root nodules on jack bean (their homologous host) under glasshouse conditions. Out of the 34 isolates, 22 formed effective nodules, and 12 formed ineffective nodules. Out of the 45 isolates, 11 were nodule endophytes or contaminants that failed to nodulate jack bean. As reported by [22], the principal component analysis (PCA) results show that soil variables correlated with the underlying microsymbiont diversity between locations. The levels of Ca, K, Mg and Fe in soils strongly influenced the distribution of microsymbionts collected from Malkerns, while the isolate from Luve was more influenced by soil pH, Zn and Cu.

### 3.1. Nodule Number and Nodule Dry Matter per Plant

All the 22 test isolates derived from jack bean produced effective nodules with a pinkish internal colour (Figure 1A). The isolates elicited variable (*p* ≤ 0.05) nodule number and nodule dry matter (NDM) values in jack bean as the homologous host (Table 2). Isolates TUTCEeS5, TUTCEeS2, TUTCEeS1, TUTCEeS8, TUTCEeS14, TUTCEeS16, TUTCEeS6 and TUTCEeS13 elicited the highest numbers of nodules, forming 195, 194, 194, 187, 181,172, 166 and 164 nodules per plant, respectively; these were followed by isolates TUTCEeS7 and TUTCEeS15. Isolate TUTCEeS21, however, produced the lowest number of nodules per plant (eight nodules per plant). Of the 22 isolates, 13 produced significantly more nodules on jack bean than the commercial *Bradyrhizobium* strain XS21, with 1 isolate (TUTCEeS3) inducing a similar nodule number as the commercial strain, and the remaining 8 isolates producing fewer nodules compared to *Bradyrhizobium* strain XS21. Generally, a high nodule number was accompanied by high nodule dry matter, and vice versa. For example, isolate TUTCEeS2 produced more nodules with higher nodule dry matter (Table 2). Similarly, isolate TUTCEeS21 produced the lowest nodule number and hence the least nodule dry mass. Six jack bean isolates (namely, TUTCEeS14, TUTCEeS16, TUTCEeS12, TUTCEeS6, TUTCEeS7 and TUTCEeS2) produced greater nodule dry matter than the commercial srain *Bradyrhizobium* XS21 (Table 2), with 6 isolates having similar NDM values as the inoculant strain XS21, and the remaining 10 isolates recording lower NDM than the commercial strain. The ratios of NDM to nodule number differed significantly (*p* ≤ 0.05) among the jack bean isolates, with values ranging from 0.82 mg.plant^−1^ for isolate TUTCEeS8 to 4.90 mg.plant^−1^ for isolate TUTCEeS17, which produced only 20 nodules.

### 3.2. Shoot Dry Matter

The shoot dry matter (SDM) yield varied (*p* ≤ 0.05) among the isolates used to inoculate jack bean seedlings (Table 2; Figure 1B). The 5 mM NO_3_-fed plants showed the highest shoot biomass (7.62 g.plant^−1^), followed by plants inoculated with isolates TUTCEeS2, TUTCEeS12, and TUTCEeS1, which recorded shoot dry matter levels of 5.78, 5.20 and 5.08 g.plant^−1^, respectively. The uninoculated control plants together with isolates TUTCEeS10, TUTCEeS11, TUTCEeS19, TUTCEeS21, TUTCEeS36 and TUTCEeS38 recorded the lowest shoot dry matter (Table 2). The isolates that induced higher nodule numbers also recorded greater shoot biomass, and vice versa (Table 3). Only 1 isolate (TUTCEeS2) recorded a greater shoot dry matter yield than the commercial *Bradyrhizobium* strain XS21, while 13 isolates showed similar SDM values as the commercial *Bradyrhizobium* strain XS21, and the remaining 8 produced lower SDM values than the commercial strain (Table 2).

The root dry matter (RDM) yield differed (*p* ≤ 0.05) among the jack bean plants inoculated with the different isolates. The 5 mM NO_3_-fed plants recorded the highest root dry matter yield (1.42 g.plant^−1^). The uninoculated control plants together with 95% of the test isolates produced similar RDM yields as the commercial *Bradyrhizobium* strain XS21, with values ranging from 0.42 to 0.67 g.plant^−1^.

The ratio of RDM to SDM (root-to-shoot ratio) varied significantly (*p* ≤ 0.05), and ranged from 0.10 g.plant^−1^ for isolate TUTCEeS12 to 0.25 g.plant^−1^ in the uninoculated control plants.

As shown in Table 2, the 5 mM NO_3_-fed plants had the highest rate of whole-plant dry matter (WPDM) accumulation (9.03 g.plant^−1^), followed by isolates TUTCEeS2, TUTCEeS12 and TUTCEeS1 (5.68 to 6.40 g.plant^−1^). The uninoculated control plants together with isolates TUTCEeS10, TUTCEeS11, TUTCEeS17, TUTCEeS21, TUTCEeS36 and TUTCEeS38 recorded the lowest WPDM yields (Table 2). Typically, the isolates that recorded high SDM also recorded high WPDM, and vice versa. For example, isolates TUTCEeS2, TUTCEeS12 and TUTCEeS1, which recorded high SDM yields, also had high WPDM, while isolates TUTCEeS36 and TUTCEeS11, which recorded low SDM, produced the lowest WPD yield (Table 2).

### 3.3. Photosynthetic Rates, Stomatal Conductance, Leaf Transpiration and WUE Induced by Test Isolates

Gas exchange measurements were performed on jack bean plants inoculated with 19 of the 22 jack bean isolates. The results showed significant differences (*p* ≤ 0.05) in photosynthetic rates (A), stomatal conductance (gs), leaf transpiration (E) and WUE (Table 3). Isolate TUTCEeS11 induced the highest photosynthetic rates (18.81 mol (CO_2_) m^−2^s^−1^), followed by isolates TUTCEeS14 and TUTCEeS2, which elicited photosynthetic rates of 14.84 mol (CO_2_) m^−2^s^−1^ and 15.00 mol (CO_2_) m^−2^s^−1^, respectively. As expected, the uninoculated control plants recorded the lowest photosynthetic rates, followed by isolate TUTCEeS36 (Table 3).

Of the 19 isolates tested, 5 elicited photosynthetic rates higher than the commercial *Bradyrhizobium* strain XS21, and 6 were similar to the commercial strain, while the remaining 8 isolates induced lower photosynthetic rates compared to the commercial strain XS21. Furthermore, 47% of the jack bean isolates elicited greater photosynthetic rates than the 5 mM NO_3_-fed plants, and two produced similar rates as the 5 mM NO_3_-fed plants (Table 3). Generally, the isolates that induced higher photosynthetic rates also elicited higher stomatal conductance and leaf transpiration, and vice versa. For example, the higher photosynthesis induced by isolates TUTCEeS11, TUTCEeS14 and TUTCEeS4 was accompanied by greater stomatal conductance and leaf transpiration. In contrast, isolates TUTCEeS13, TUTCEeS8, TUTCEeS36 and TUTCEeS38 induced lower photosynthetic rates, with reduced stomatal conductance and leaf transpiration, which were expectedly associated with high WUE (estimated as the ratio of photosynthesis to stomatal conductance) (Table 3).

### 3.4. Relative Symbiotic Effectiveness

The jack bean isolates revealed marked differences (*p* ≤ 0.05) in relative symbiotic effectiveness (Table 2). Isolate TUTCEeS2 recorded the highest RSE (124.64%), followed by isolates TUTCEeS12 and TUTCEeS1 (112.07 and 109.55% RSE, respectively). In contrast, isolates TUTCEeS10, TUTCEeS11 and TUTCEeS36 showed much lower RSE values, which ranged from 54.45 to 62.14%. Only isolate TUTCEeS2 exhibited a much higher RSE than the commercial *Bradyrhizobium* strain XS21, with 59% of isolates recording similar RSE values to the commercial strain XS21. About 36% of the isolates produced lower RSE values than the commercial strain XS21. Fifteen isolates were highly effective, and recorded RSE > 80%. The remaining seven isolates with RSE of 50 to 80% were regarded as moderate N_2_-fixers.

### 3.5. Symbiotic Parameters

#### 3.5.1. Shoot %N, N Content, δ ^15^N and N-Fixed

Isolate TUTCEeS6 produced the highest shoot %N, followed by TUTCEeS16, TUTCEeS14 and TUTCEeS7 (Table 4). However, the uninoculated plants, 5 mM NO_3_-fed plants, and two jack bean isolates (TUTCEeS19 and TUTCEeS21) produced the lowest shoot %N. Sixteen isolates produced shoot N contents similar to that of the commercial *Bradyrhizobium* strain XS21 (Table 4). In contrast, the uninoculated control plants, 5 mM NO_3_-fed plants and isolates TUTCEeS9, TUTCEeS11, TUTCEeS17, TUTCEeS19, TUTCEeS21 and TUTCEeS36 revealed much lower shoot N contents than the commercial *Bradyrhizobium* strain XS21 (Table 4).

Generally, isolates that recorded high shoot %N also recorded high N content, and vice versa. For example, isolates TUTCEeS6 and TUTCEeS14 produced high shoot %N and high shoot N content, while isolates TUTCEeS19 and TUTCEeS21 induced low shoot %N and N content. Furthermore, the isolates that recorded high SDM, shoot %N and N content consistently revealed low shoot δ^15^N values, and vice versa, with few exceptions (Table 4). Isolates TUTCEeS1 and TUTCEeS12 with lower δ^15^N values, for example, induced high SDM, shoot %N and N content, in contrast to isolates TUTCEeS11 and TUTCEeS36, with higher δ^15^N values and lower SDM, shoot %N and shoot N content (Table 4). Moreover, the differences in N contents between the non-inoculated and inoculated plants and the %RSE showed positive correction (r = 0.8548) (Figure 2).

The amount of N-fixed varied significantly (*p* ≤ 0.05) between and among the test isolates. For example, isolate TUTCEeS2 contributed the highest amount of N, followed by TUTCEeS6, TUTCEeS12, TUTCEeS16 and TUTCEeS7 (Table 4), with isolate TUTCEeS36 producing the least amount of symbiotic N. Thirteen of the isolates fixed similar amounts of symbiotic N as the commercial *Bradyrhizobium* strain XS21, while nine produced less N than the commercial strain XS21. It is noteworthy that, consistently, all the isolates that recorded high shoot δ^15^N values also produced low amounts of N-fixed, and vice versa (Table 4). For example, isolates TUTCEeS1, TUTCEeS2, TUTCEeS3 and TUTCEeS4, which revealed low shoot δ^15^N values, exhibited higher amounts of N-fixed, while isolates TUTCEeS9, TUTCEeS11, TUTCEeS21 and TUTCEeS36 recorded greater shoot δ^15^N values and low amounts of N-fixed. Furthermore, greater N-fixed resulted in high SDM accumulation (Table 4).

#### 3.5.2. Root %N, N Content, δ ^15^N and N-Fixed

As with shoot N concentration, isolate TUTCEeS6 induced the highest root N concentration, followed by TUTCEeS18, with TUTCEeS19 producing the lowest root N concentration (Table 4). Isolates TUTCEeS6, TUTCEeS18, TUTCEeS15, TUTCEeS1 and TUTCEeS13 elicited a significantly greater root N concentration than the commercial strain XS21, with 10 isolates inducing similar root N concentrations as the commercial *Bradyrhizobium* strain XS21. The uninoculated control plants, 5 mM NO_3_-fed plants, and seven isolates recorded significantly lower root N concentrations than the commercial strain XS21 (Table 4). A similar trend was also observed, whereby high root %N resulted in high root N content, and vice versa. Isolates TUTCEeS1 and TUTCEeS6, for example, recorded high root %N leading to high root N content, while isolates TUTCEeS11 and TUTCEeS17 recorded low root %N and low root N content. The isolates that recorded high root %N and root N content generally had low δ^15^N values, and vice versa. For example, isolates TUTCEeS1 and TUTCEeS15 exhibited high root %N and N content with lower δ^15^N values, while isolates TUTCEeS21 and TUTCEeS36 recorded lower root %N and N content with higher δ^15^N values.

There were marked differences (*p* ≤ 0.05) in the amount of N-fixed by the 22 jack bean isolates (Table 4). For example, isolate TUTCEeS6 contributed the highest symbiotic N, and TUTCEeS38 the least. Only isolate TUTCEeS6 fixed more N than the commercial *Bradyrhizobium* strain XS21 (Table 4). The remaining 21 isolates produced similar amounts of symbiotic N to the commercial strain XS21 (Table 4).

#### 3.5.3. Whole-Plant %N, N Content, δ ^15^N and N-Fixed

The whole-plant N concentration was greater with isolate TUTCEeS6, followed by TUTCEeS16, TUTCEeS14, TUTCEeS7 and TUTCEeS13 (Table 4). The uninoculated control plants and 5 mM NO_3_-fed plants, together with isolates TUTCEeS13, TUTCEeS21, TUTCEeS11 and TUTCEeS36, produced the least whole-plant N (Table 4). As with shoot and root %N, the isolates that produced the highest whole-plant %N also produced high N content, and vice versa. For example, isolates TUTCEeS2, TUTCEeS5 and TUTCEeS6 produced high whole-plant %N, and hence high N content, while isolates TUTCEeS13, TUTCEeS21 and TUTCEeS11 recorded low whole-plant %N and therefore a low N content (Table 4). It is interesting to note that the isolates that recorded high WPDM, whole-plant %N and N content also had low δ^15^N values, and vice versa. For example, isolates TUTCEeS1, TUTCEeS2 and TUTCEeS6 recorded low δ^15^N values and hence high WPDM, whole-plant %N and N content, while isolates TUTCEeS36, TUTCEeS21 and TUTCEeS11 with high δ^15^N values recorded lower WPDM, whole-plant %N and N content (Table 4).

The amount of N-fixed at the whole-plant level varied (*p* ≤ 0.05) among the 22 jack bean isolates (Table 4), with TUTCEeS2 contributing the highest symbiotic N, followed by isolate TUTCEeS6, and TUTCEeS36 the least (Table 4). Thirteen of the isolates could fix similar amounts of N to the commercial strain, while the remaining nine produced less N compared to the commercial *Bradyrhizobium* strain XS21 (Table 4). Similar to the shoot data, all the isolates that produced high δ^15^N values at whole-plant level also recorded less N-fixed, and vice versa. Isolates TUTCEeS21 and TUTCEeS36, for example, recorded high δ^15^N values at the whole-plant level and fixed less N, while isolates TUTCEeS1 and TUTCEeS7 produced lower δ^15^N values and fixed higher amounts of N. High N-fixed also led to high WPDM accumulation.

### 3.6. Carbon Assimilation and δ^13^C Values

#### 3.6.1. Shoot %C, C Content, C:N Ratio and δ^13^C

There were marked differences (*p* ≤ 0.05) in the shoot %C of the test isolates (Table 5), with TUTCEeS12 recording the highest shoot %C (43.88%), followed by TUTCEeS14 (43.68%), and the uninoculated control plants together with isolate TUTCEeS19 (40.22% and 40.51%, respectively) showed the lowest shoot %C (Table 5). Isolate TUTCEeS12 also had a much higher shoot %C when compared to the commercial *Bradyrhizobium* strain XS21, while the 5 mM NO_3_-fed plants together with plants inoculated by 15 isolates recorded shoot %C values similar to the commercial *Bradyrhizobium* strain XS21. Only isolates TUTCEeS9, TUTCEeS11, TUTCEeS17, TUTCEeS19, TUTCEeS21 and TUTCEeS36 recorded %C values lower than the commercial *Bradyrhizobium* strain XS21 (Table 5).

The 5 mM NO_3_-fed plants recorded the highest shoot C content (3300.42 mg.plant^−1^), followed by isolates TUTCEeS2 and TUTCEeS12, with 2500.50 mg.plant^−1^ and 2282.06 mg.plant^−1^, respectively (Table 5). The uninoculated control plants together with isolates TUTCEeS36, TUTCEeS11, TUTCEeS19 and TUTCEeS10 produced the lowest shoot C content. Isolate TUTCEeS2 recorded greater C content than the commercial *Bradyrhizobium* strain XS21 (Table 5). However, 14 of the test isolates had shoot C contents similar to the commercial strain XS21, and 7 isolates revealed shoot C contents lower than the commercial strain.

The highest C:N ratio was recorded by the 5 mM NO_3_-fed plants (30.42), followed by the uninoculated control plants (28.70), and the least was produced by isolate TUTCEeS6 (11.44) (Table 5). The 5 mM NO_3_-fed plants and the uninoculated control plants, together with 6 isolates (namely, TUTCEeS9, TUTCEeS11, TUTCEeS17, TUTCEeS19, TUTCEeS21 and TUTCEeS36), recorded C:N ratios that were higher than the commercial strain XS21, and 16 showed C:N values similar to the commercial strain XS21 (Table 5).

The shoot δ^13^C values of the isolates varied significantly (*p* ≤ 0.05), with TUTCEeS8 recording the highest (−23.87‰), followed by TUTCEeS1 (−23.98‰) (Table 5). The uninoculated control plants recorded the lowest shoot δ^13^C values (−26.08‰). Seven isolates (namely, TUTCEeS5, TUTCEeS1, TUTCEeS15, TUTCEeS13, TUTCEeS6 and TUTCEeS2) recorded greater shoot δ^13^C values than the commercial strain XS21. However, the 5 mM NO_3_-fed plants, together with 13 isolates, revealed shoot δ^13^C values similar to that of the commercial *Bradyrhizobium* strain XS21 (Table 5), while the uninoculated control plants, together with isolates TUTCEeS21 and TUTCEeS11, recorded shoot δ^13^C values lower than the commercial strain XS21 (Table 5).

#### 3.6.2. Root %C, C Content, C:N Ratio and δ^13^C

The root %C ranged from 39.02% with isolate TUTCEeS12 to 45.06% for isolate TUTCEeS8 (Table 5). The 5 mM NO_3_-fed plants and the uninoculated control plants, together with 21 test isolates, recorded similar root %C values to those of the commercial *Bradyrhizobium* strain XS21, with isolate TUTCEeS8 showing a lower root %C than the commercial strain (Table 5).

The 5 mM NO_3_-fed plants recorded the highest root C content. However, there were no marked differences (*p* = 0.05) in the C contents of the uninoculated control plants, the 22 jack bean isolates and the commercial *Bradyrhizobium* strain XS21 (Table 5).

Isolates TUTCEeS11 and TUTCEeS19 recorded the highest root C:N ratio (24.88 and 23.92, respectively), followed by isolate TUTCEeS38 (23.77), while TUTCEeS6 produced the lowest C:N ratio (15.78). The 5 mM NO_3_-fed plants and the uninoculated control plants, together with 6 test isolates, recorded C:N ratios higher than the commercial strain XS21, while 13 isolates had C:N ratios similar to the commercial strain XS21 (Table 5). Isolates TUTCEeS6, TUTCEeS13 and TUTCEeS18 produced C:N values that were lower than that of the commercial strain XS21 (Table 5).

The root δ^13^C values varied (*p* ≤ 0.05) between and among the jack bean isolates (Table 5), with TUTCEeS18 recording the highest root δ^13^C (−22.73‰), followed by isolates TUTCEeS9, TUTCEeS5 and TUTCEeS6 (−23.05‰, −23.10‰ and −23.11‰, respectively) and the 5 mM NO_3_-fed plants (−23.07‰). The uninoculated control plants, together with isolate TUTCEeS10, registered the lowest root δ^13^C values (−24.86‰ and −24.97‰, respectively). The 5 mM NO_3_-fed plants, together with 11 isolates, produced root δ^13^C values that were higher than the commercial strain XS21, while 10 isolates recorded root δ^13^C values that were similar to the commercial strain XS21 (Table 5). The uninoculated control plants together with isolate TUTCEeS10 had root δ^13^C values that were lower than the commercial *Bradyrhizobium* strain XS21 (Table 5).

## 4. Discussion

### 4.1. Symbiotic Effectiveness and Its Effect on Photosynthetic Rates Induced by Microsymbionts Nodulating Jack Bean in Eswatini Soils

Twenty-two rhizobial isolates of jack bean from various locations in Eswatini were assessed for symbiotic effectiveness under glasshouse conditions. The isolates varied in symbiotic functioning, measured as nodulation, plant growth and photosynthetic rates. Nodule numbers ranged from 8 per plant for TUTCEeS21 to 195 per plant for TUTCEeS5, with an increase in nodule numbers generally resulting in higher nodule dry matter, as confirmed by the significant correlation found between nodule number and nodule dry matter in this study (Figure 3A) and other studies involving Bambara groundnut [33]. However, an increase in effective nodule number can also correlate with shoot biomass, as found in this study (Figure 3B,C). This implies that the enhanced N_2_ fixation caused by the rhizobia in root nodules increased C accumulation via photosynthesis, affecting plant growth and biomass accumulation in jack bean plants. Our argument is consistent with the observed relationship between nodulation and shoot biomass in Bambara groundnut and Kersting’s groundnut [32,33]. In this study, however, nodule dry matter was also directly correlated with photosynthetic rates, clearly indicating a link between nodule function and photosynthesis in nodulated jack bean (Figure 3D). As found for Bambara groundnut [33], an increase in stomatal conductance induced by the applied rhizobial isolates in this study were generally accompanied by greater photosynthetic rates in jack bean (Figure 3E,F), leading to increased shoot and whole-plant dry matter accumulation (Figure 4A and Figure 6A).

This study also assessed the percent relative symbiotic effectiveness (%RSE) of the jack bean microsymbionts, which altered the photosynthetic functioning of the host plant. The values relative symbiotic effectiveness of jack bean isolates were found to vary significantly, and ranged from 54.5% for isolate TUTCEeS36 to 124.6% for TUTCEeS2. Seven jack bean isolates showed RSE values that were higher than the commercial *Bradyrhizobium* strain XS21, suggesting that the indigenous rhizobia present in Malkerns soils have the potential to be used in inoculant formulations. In fact, several studies have reported the presence of highly effective native rhizobia in African soils when compared to the commercial rhizobial inoculants currently in use [32,34,49].

### 4.2. Symbiotic Performance and C Assimilation by the 22 Microsymbionts Nodulating Jack Bean in Eswatini

The 22 jack bean isolates were variable in their levels of N_2_ fixation (measured as plant %N concentration and amounts of N-fixed) at the shoot, root and whole-plant levels (Table 4). Generally, high N_2_-fixing effectiveness, measured as increased N concentration, N content and amount of N-fixed in plant parts, resulted in high shoot, root and whole-plant dry matter accumulation (Table 4). This observation was reinforced by the strong positive correlation found between shoot, root and whole-plant dry matter vs. N concentration and amount of N-fixed (Figure 4B,C, Figure 5A,B and Figure 6B,C). Similar findings have been reported [15]. In fact, 59% of the jack bean isolates fixed similar amounts on N as the commercial *Bradyrhizobium* strain XS21 (Table 4), at both the shoot and whole-plant levels, thus confirming the potential of these strains for use as inoculants. What was also interesting to note is that the jack bean isolates that recorded high %RSE also fixed greater amounts of N at the shoot and whole-plant levels (Table 2 and Table 4).

Furthermore, the high biomass at the shoot and root levels was the consequence of increased C accumulation resulting from photosynthesis (Table 5), hence the strong positive relationship between shoot or root dry matter and C content (Figure 4D and Figure 5C). The significant correlation between shoot dry matter and δ^13^C (Figure 4E) suggests that jack bean inoculation with rhizobia can promote plant growth and increase water use efficiency. Furthermore, 17 out of the 22 jack bean isolates recorded C:N values of 9.4 to 22.7 g g^−1^. Refs. [50,51], however, reported that plant residues of legumes with C:N ratios of 9.4 to 22.7 g g^−1^ usually undergo faster mineralization. This therefore suggests that if this is incorporated into the soil, jack bean plant residues can be decomposed faster, thus improving the fertility of the soil.

In conclusion, inoculating jack bean with effective rhizobial isolates from diverse locations in Eswatini resulted in increased nodulation, shoot biomass accumulation, photosynthetic rates and symbiotic performance. Seven isolates (TUTCEeS2, TUTCEeS12, TUTCEeS1, TUTCEeS17, TUTCEeS6, TUTCEeS18, TUTCEeS5) in this study recorded greater percentages of relative symbiotic effectiveness than the commercial strain XS21, and therefore have potential to be used in inoculant formation after field testing for competitiveness. Furthermore, all the isolates that recorded high %RSE values also produced high amounts of N-fixed, which again attests to the potential of these strains for use in inoculant manufacture. We do, however, recommend the field testing of these strains for their competitiveness with native soil rhizobia before their consideration for use in inoculant production.

## Figures and Tables

**Figure 1 microorganisms-11-02786-f001:**
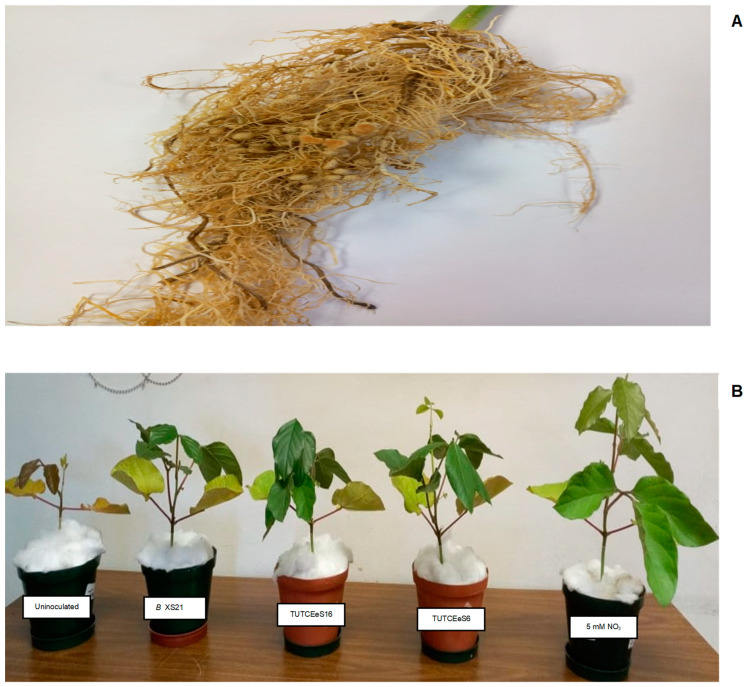
(**A**) Nodulated root of jack bean showing effective nodules. (**B**) A comparison of jack bean plant growth caused by different rhizobial isolates and the commercial strain.

**Figure 2 microorganisms-11-02786-f002:**
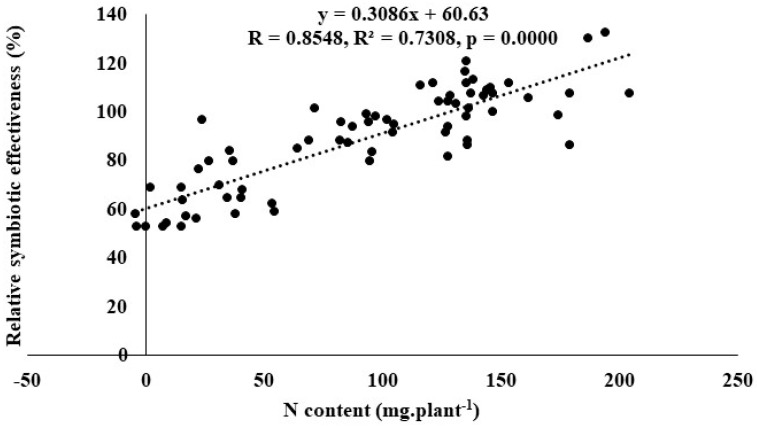
Correlation and regression analysis between N content and relative symbiotic effectiveness of jack bean inoculated with rhizobia from diverse locations (Luve, KaZulu and Malkerns Research Station) in Eswatini.

**Figure 3 microorganisms-11-02786-f003:**
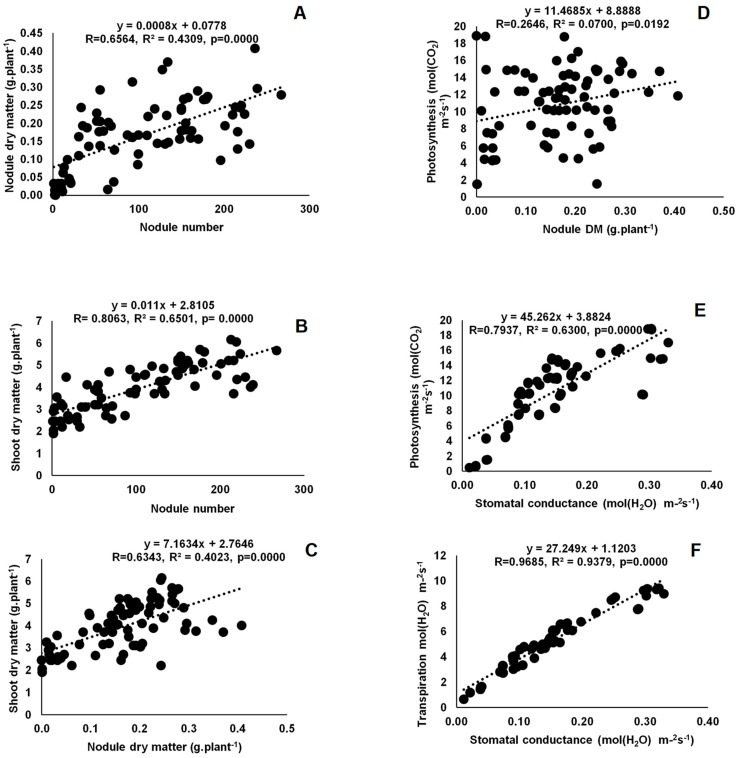
Correlation and regression analysis between (**A**) nodule number and nodule biomass, (**B**), nodule number and shoot dry matter, (**C**) nodule biomass and shoot dry matter, (**D**) nodule and photosynthesis, (**E**) stomatal conductance and photosynthesis and (**F**) stomatal conductance and transpiration of jack bean inoculated with rhizobia from diverse locations (Luve, KaZulu and Malkerns Research Station) in Eswatini.

**Figure 4 microorganisms-11-02786-f004:**
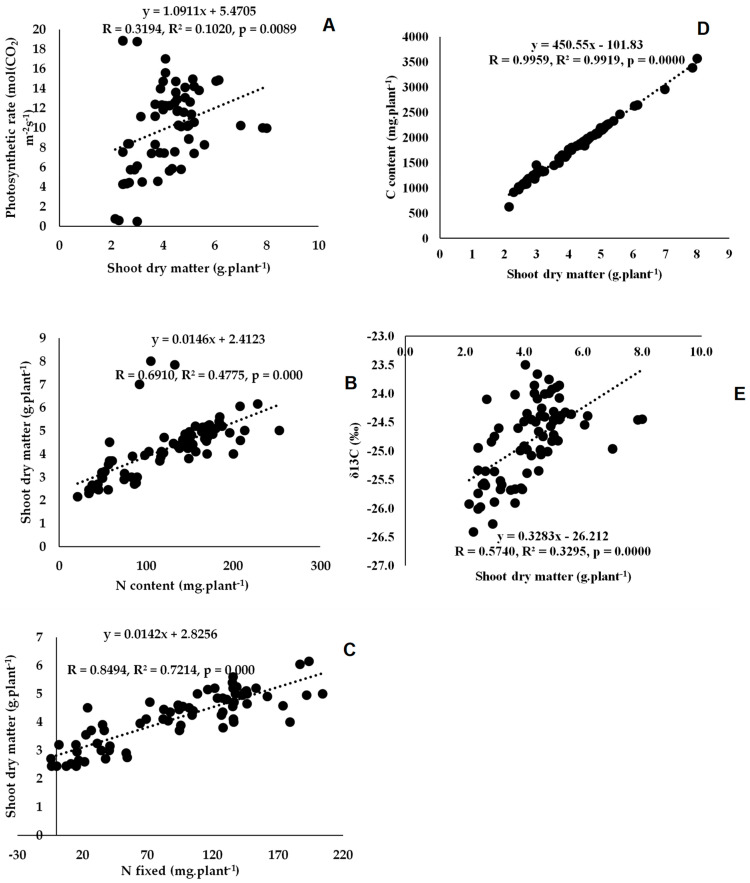
Correlation and regression analysis between (**A**) shoot dry matter and photosynthetic rate, (**B**) shoot dry matter and N content, (**C**) shoot dry matter and N fixation, (**D**) shoot dry matter and C content and (**E**) shoot dry matter and shoot δ^13^C of jack bean inoculated with rhizobia from diverse locations (Luve, KaZulu and Malkerns Research Station) in Eswatini.

**Figure 5 microorganisms-11-02786-f005:**
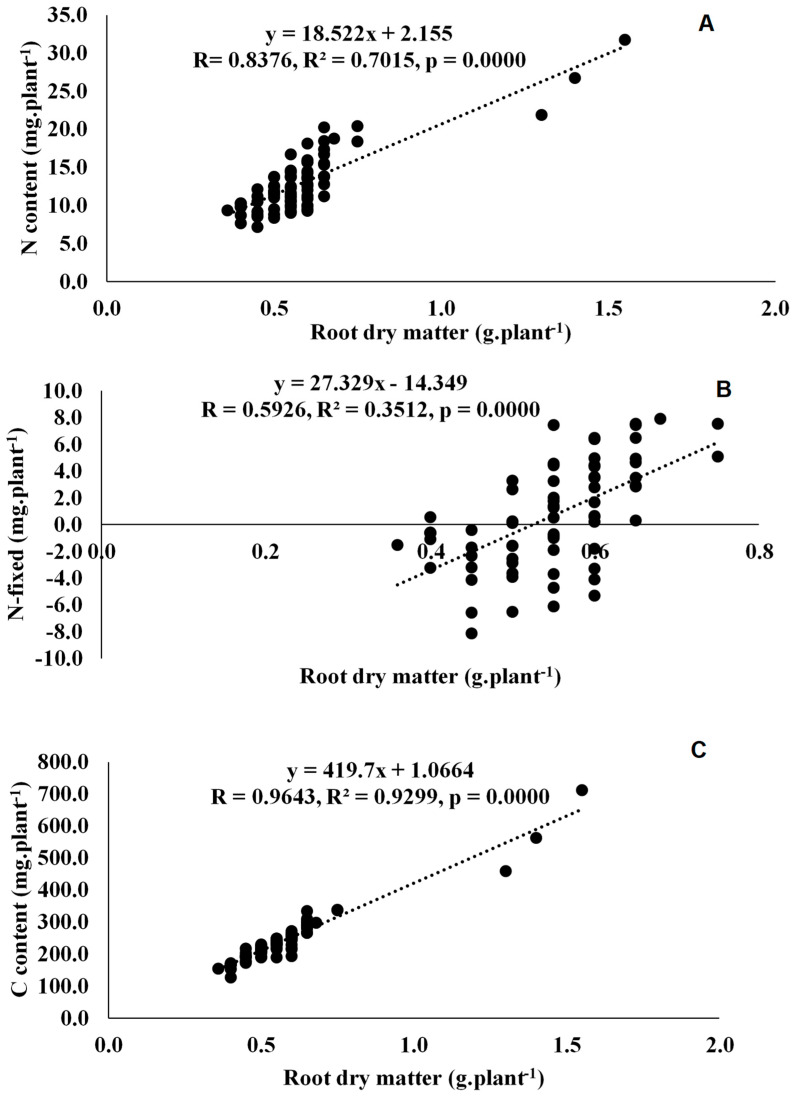
Correlation and regression analysis between (**A**) root dry matter and root N content, (**B**) root dry matter and root N-fixed and (**C**) root dry matter and root C content of jack bean inoculated with rhizobia from diverse locations (Luve, KaZulu and Malkerns Research Station) in Eswatini.

**Figure 6 microorganisms-11-02786-f006:**
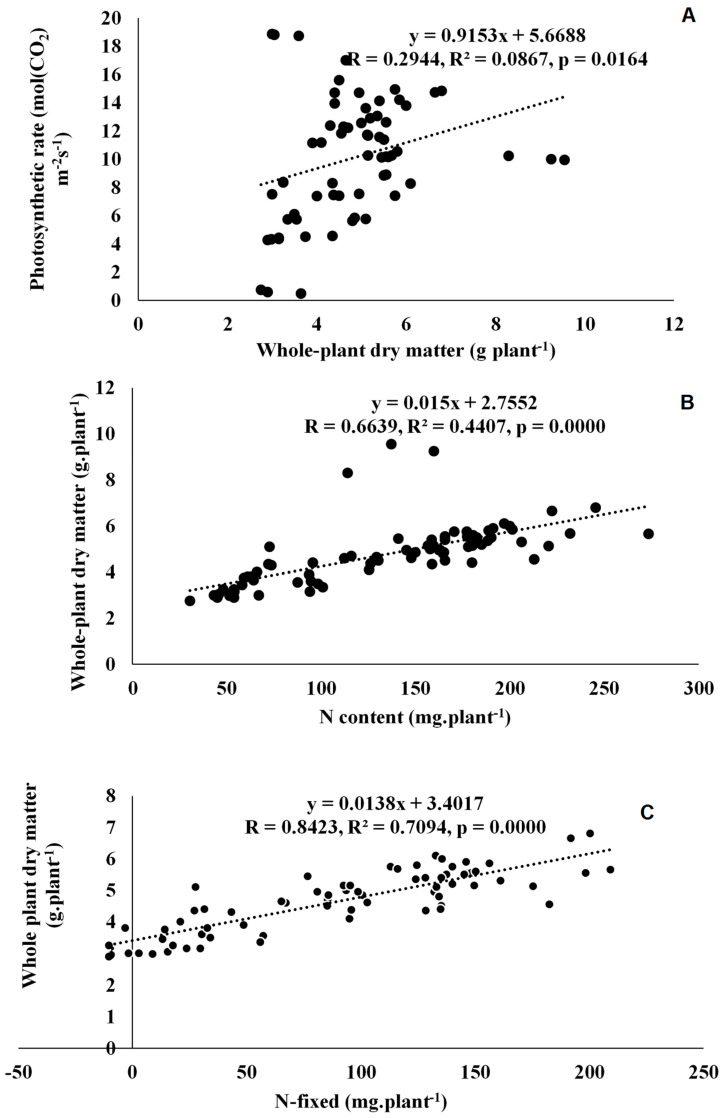
Correlation and regression analysis between (**A**) photosynthetic rate and whole-plant DM, (**B**) photosynthetic rate and whole-plant N content and (**C**) whole-plant DM and whole-plant N fixation of jack bean inoculated with rhizobia from diverse locations (Luve, KaZulu and Malkerns Research Station) in Eswatini.

**Table 1 microorganisms-11-02786-t001:** Chemical properties of bulk soil from Ka-Zulu, Luve and New Heaven farmers’ fields, and Malkerns Research Station for the 2016/2017 season [22].

Nutrient	Ka-Zulu Farm	New Heaven	Luve	Malkerns
pH (KCl)	4.3	3.9	6.1	4.4
Total N (%)	0.069	0.112	0.036	0.075
Available P (mg·kg^−1^)	4	13	53	14
K (mg·kg^−1^)	25	41	21	54
Cu (mg·kg^−1^)	1.47	5.9	0.14	0.91
Zn (mg·kg^−1^)	2.17	1.68	2.09	1.69
Ca (mg·kg^−1^)	0.22	1.09	0.35	1.15
Fe (mg·kg^−1^)	54.69	152.1	81.66	452.80
Mg cmol(+)Fe (mg·kg^−1^)	0.13	0.81	0.18	0.84
Na (mg·kg^−1^)	7	25	9	5

**Table 2 microorganisms-11-02786-t002:** Nodulation, plant growth and relative symbiotic effectiveness of jack bean isolates from various locations in Eswatini; the plants were harvested 60 days after planting. Values (Mean ± SE) of dissimilar letters in a column are significantly different at *p* ≤ 0.05, *** *p* ≤ 0.001, NA = not applicable.

Isolate	Location	Nodule Number	Nodule Dry Matter	Nodule Dry Matter/Nodule Number	Shoot Dry Matter	Root Dry Matter	Root Dry Matter/Shoot Dry Matter	Whole-Plant Dry Matter	Relative Symbiotic Effectiveness
			g.plant^−1^	mg. plant^−1^	g.plant^−1.^	g.plant^−1^	g.plant^−1^	g.plant^−1^	%
TUTCEeS1	Malkerns	194.00 ± 9.49 a	0.20 ± 0.01 de	1.04 ± 0.09 gh	5.08 ± 0.12 bc	0.60 ± 0.03 bc	0.12 ± 0.00 g–k	5.68 ± 0.14 bc	109.55 ± 2.51 ab
TUTCEeS2	Malkerns	194.00 ± 22.07 a	0.24 ± 0.00 bc	1.29 ± 0.16 fgh	5.78 ± 0.32 b	0.62 ± 0.02 b	0.11 ± 0.01 ijk	6.40 ± 0.33 b	124.64 ± 6.85 a
TUTCEeS3	Malkerns	126.00 ± 3.46 ef	0.14 ± 0.01 gh	1.09 ± 0.07 gh	3.77 ± 0.38 gh	0.57 ± 0.10 bcd	0.16 ± 0.01 c–h	4.33 ± 0.34 fg	81.18 ± 8.12 ef
TUTCEeS4	Malkerns	56.00 ± 2.65 g	0.13 ± 0.01 ghi	2.30 ± 0.28 c	4.05 ± 0.05 efg	0.60 ± 0.03 bc	0.15 ± 0.01 d–j	4.65 ± 0.03 def	87.28 ± 1.08 de
TUTCEeS5	Malkerns	195.00 ± 38.35 a	0.20 ± 0.01 de	1.09 ± 0.19 gh	4.82 ± 0.15 c–f	0.55 ± 0.05 bcd	0.12 ± 0.01 g–k	5.37 ± 0.10 cde	103.81 ± 3.13 bcd
TUTCEeS6	Malkerns	166.00 ± 6.33 a–d	0.25 ± 0.01 b	1.51 ± 0.20 efg	4.87 ± 0.27 cd	0.63 ± 0.02 b	0.13 ± 0.00 f–k	5.50 ± 0.28 c	104.89 ± 5.78 bc
TUTCEeS7	Malkerns	157.00 ± 6.89 b–e	0.24 ± 0.01 bc	1.52 ± 0.08 efg	4.65 ± 0.14 c–f	0.45 ± 0.03 cd	0.10 ± 0.01 k	5.10 ± 0.13 c–f	100.22 ± 3.11 bcd
TUTCEeS8	Malkerns	187.00 ± 6.51 ab	0.15 ± 0.01 fg	0.82 ± 0.05 h	4.57 ± 0.34 c–f	0.50 ± 0.03 bcd	0.11 ± 0.00 h–k	5.07 ± 0.37 c–f	98.42 ± 7.26 bcd
TUTCEeS9	Malkerns	141.00 ± 11.50 de	0.12 ± 0.01 hi	0.83 ± 0.03 h	4.03 ± 0.24 fg	0.57 ± 0.03 bcd	0.14 ± 0.01 e–k	4.60 ± 0.25 ef	86.93 ± 5.18 de
TUTCEeS10	Malkerns	20.00 ± 1.76 l–p	0.06 ± 0.01 j	3.23 ± 0.11 b	2.88 ± 0.07 ij	0.58 ± 0.04 bc	0.20 ± 0.02 b–f	3.47 ± 0.06 h	62.14 ± 1.57 g
TUTCEeS11	Malkerns	48.00 ± 3.38 gh	0.11 ± 0.01 hi	2.23 ± 0.05 cd	2.63 ± 0.18 ij	0.58 ± 0.02 bc	0.22 ± 0.01 ab	3.22 ± 0.19 h	56.75 ± 3.95 g
TUTCEeS12	Malkerns	149.00 ± 5.90 cde	0.25 ± 0.01 b	1.68 ± 0.02 d–g	5.20 ± 0.20 bc	0.52 ± 0.02 bcd	0.10 ± 0.01 jk	5.72 ± 0.19 bc	112.07 ± 4.31 ab
TUTCEeS13	Malkerns	164.00 ± 8.72 a–d	0.21 ± 0.01 cd	1.30 ± 0.13 fgh	4.43 ± 0.14 c–g	0.48 ± 0.04 bcd	0.11 ± 0.01 h–k	4.92 ± 0.09 c–f	95.55 ± 2.94 b–e
TUTCEeS14	Malkerns	181.00 ± 9.74 abc	0.30 ± 0.01 a	1.66 ± 0.09 d–g	4.20 ± 0.15 d–g	0.42 ± 0.02 d	0.10 ± 0.00 jk	4.62 ± 0.17 ef	90.52 ± 3.29 cde
TUTCEeS15	Malkerns	155.00 ± 4.04 b–e	0.17 ± 0.01 ef	1.11 ± 0.03 gh	4.92 ± 0.12 cd	0.60 ± 0.10 bc	0.12 ± 0.02 g–k	5.52 ± 0.04 c	105.96 ± 2.51 bc
TUTCEeS16	Malkerns	172.00 ± 14.19 a–d	0.28 ± 0.01 a	1.66 ± 0.06 d–g	4.55 ± 0.43 c–f	0.50 ± 0.07 bcd	0.11 ± 0.02 g–k	5.05 ± 0.47 c–f	98.06 ± 9.29 bcd
TUTCEeS17	Malkerns	20.00 ± 15.34 hi	0.10 ± 0.01 i	4.90 ± 0.49 a	3.29 ± 0.43 hi	0.50 ± 0.03 bcd	0.16 ± 0.03 c–h	3.79 ± 0.41 gh	70.98 ± 9.32 fg
TUTCEeS18	Malkerns	139.00 ± 8.69 de	0.23 ± 0.01 bcd	1.64 ± 0.11 d–g	4.85 ± 0.13 cde	0.61 ± 0.04 b	0.13 ± 0.01 g–k	5.46 ± 0.16 cd	104.53 ± 2.71 bc
TUTCEeS19	Malkerns	36.00 ± 1.20 hi	0.11 ± 0.01 hi	3.15 ± 0.19 b	3.02 ± 0.34 hij	0.60 ± 0.03 bc	0.20 ± 0.01 bc	3.62 ± 0.37 gh	65.01 ± 7.37 fg
TUTCEeS21	Malkerns	8.00 ± 0.67 i	0.01 ± 0.00 k	1.90 ± 0.48 c–f	3.13 ± 0.09 hij	0.55 ± 0.03 bcd	0.18 ± 0.01 b–f	3.68 ± 0.12 gh	67.53 ± 2.00 fg
TUTCEeS36	New Heaven	17.00 ± 1.73 hi	0.03 ± 0.00 jk	2.08 ± 0.12 cde	2.53 ± 0.04 ij	0.48 ± 0.03 bcd	0.19 ± 0.01 bcd	3.01 ± 0.07 h	54.45 ± 0.93 g
TUTCEeS38	Luve	56.00 ± 3.93 g	0.20 ± 0.01 de	3.58 ± 0.23 b	3.23 ± 0.32 hij	0.52 ± 0.03 bcd	0.16 ± 0.01 c–g	3.75 ± 0.35 gh	69.68 ± 6.85 fg
B. XS21	-	105.00 ± 0.88 f	0.19 ± 0.02 de	1.86 ± 0.11 c–f	4.64 ± 0.08 c–f	0.58 ± 0.02 bc	0.13 ± 0.00 g–k	5.23 ± 0.09 cde	100.07 ± 1.70 bcd
5 mM KNO_3_	-	NA	NA	NA	7.62 ± 0.31 a	1.42 ± 0.07 a	0.19 ± 0.00 b–e	9.03 ± 0.38 a	NA
Uninoculated (Control)	-	NA	NA	NA	2.48 ± 0.26 j	0.62 ± 0.02 b	0.25 ± 0.02 a	3.10 ± 0.28 h	NA
F statistics	-	55.29 ***	17.89 ***	12.39 ***	21.42 ***	13.67 ***	11.13 ***	21.87 ***	15.55 ***

**Table 3 microorganisms-11-02786-t003:** Photosynthetic induction of jack bean isolates from various locations in Eswatini. Values (Mean ± SE) of dissimilar letters in a column are significantly different at *p* ≤ 0.05, *** *p* ≤ 0.001, ND = not determine.

Isolate	Photosynthic Rate	Stomatal Conductance	Transpiration Rate	Wate Use Efficiency
	mol (CO_2_) m^−2^s^−1^	mol (H_2_O) m^−2^s^−1^	mol (H_2_O) m^−2^s^−1^	µmol (CO^2^) mol^−1^ (H_2_O)
TUTCEeS1	12.96 ± 1.20 cd	0.14 ± 0.02 c–g	5.73 ± 0.83 b–e	94.87 ± 9.11 bc
TUTCEeS2	14.84 ± 0.06 b	0.14 ± 0.00 c–e	4.98 ± 0.00 def	103.41 ± 0.33 ab
TUTCEeS3	13.85 ± 0.47 b	0.18 ± 0.01 de	6.31 ± 0.22 bc	63.16 ± 0.13 ij
TUTCEeS4	11.62 ± 1.57 bc	0.21 ± 0.06 b	6.52 ± 1.23 b	72.85 ± 10.67 ghi
TUTCEeS5	12.86 ± 0.13 cd	0.18 ± 0.00 bc	6.61 ± 0.02 b	73.44 ± 0.48 f–i
TUTCEeS6	ND	ND	ND	ND
TUTCEeS7	ND	ND	ND	ND
TUTCEeS8	7.46 ± 0.05 g	0.09 ± 0.00 ghi	3.76 ± 0.01 ghi	82.50 ± 0.36 c–g
TUTCEeS9	13.31 ± 0.4 c	0.15 ± 0.01 cde	4.80 ± 0.17 efg	91.35 ± 4.95 b–e
TUTCEeS10	5.86 ± 0.12 h	0.07 ± 0.00 hij	3.30 ± 0.01 hi	79.68 ± 1.73 d–g
TUTCEeS11	18.81 ± 0.04 a	0.30 ± 0.00 a	9.30 ± 0.04 a	62.45 ± 0.40 ij
TUTCEeS12	8.68 ± 0.20 g	0.09 ± 0.00 g–i	4.20 ± 0.19 fgh	93.70 ± 5.81 bcd
TUTCEeS13	5.74 ± 5.74 hi	0.07 ± 0.00 hij	2.74 ± 0.00 i	77.83 ± 0.87 e–h
TUTCEeS14	15.00 ± 0.30 b	0.18 ± 0.02 bc	6.54 ± 0.47 b	87.45 ± 8.57 c–f
TUTCEeS15	10.56 ± 0.41 ef	0.10 ± 0.01 e–i	3.38 ± 0.23 hi	103.25 ± 6.04 ab
TUTCEeS16	12.63 ± 0.59 cd	0.15 ± 0.02 cde	5.23 ± 0.43 cf	87.76 ± 7.07 c–f
TUTCEeS17	7.46 ± 0.04 g	0.12 ± 0.00 d–h	4.89 ± 0.01 ef	60.68 ± 0.28 ij
TUTCEeS18	10.24 ± 0.03 jef	0.11 ± 0.00 d–i	4.78 ± 0.00 efg	95.47 ± 0.25 bc
TUTCEeS19	8.35 ± 0.02 g	0.15 ± 0.00 cd	5.47 ± 0.02 b–e	56.12 ± 0.24 j
TUTCEeS21	ND	ND	ND	ND
TUTCEeS36	4.32 ± 0.02 i	0.04 ± 0.00 jk	1.44 ± 0.01 j	113.44 ± 0.49 a
TUTCEeS38	4.50 ± 0.04 hi	0.07 ± 0.00 ij	2.82 ± 0.00 i	65.09 ± 0.52 hij
B. XS21	11.66 ± 0.04 de	0.11 ± 0.00 d–i	3.35 ± 0.00 hi	110.64 ± 0.43 b
5 mM KNO_3_	10.06 ± 0.09 f	0.16 ± 0.00 cd	6.04 ± 0.03 bcd	64.01 ± 0.36 ij
Uninoculated (Control)	0.61 ± 0.08 j	0.02 ± 0.00 k	0.10 ± 0.18 j	34.94 ± 4.06 k
F statistics	112.32 ***	30.71 ***	46.95 ***	31.37 ***

**Table 4 microorganisms-11-02786-t004:** Symbiotic parameters of 22 jack bean isolates sourced from various locations in Eswatini; the plants were harvested 60 days after planting. Values (Mean ± SE) of dissimilar letters in a column are significantly different at *p* ≤ 0.05, ** *p* ≤ 0.01, *** *p* ≤ 0.001, ND = not determine.

Isolate	Shoot					Root					Whole-Plant				
	DM	N	N Content	δ^15^N	N-Fixed	DM	N	N Content	δ^15^N	N-Fixed	DM	N	N Content	δ^15^N	N-Fixed
	g.plant^−1^	%	mg.plant^−1^		mg.plant^−1^	g.plant^−1^	%	mg.plant^−1^		kg.ha^−1^	g.plant^−1^	%	mg.plant^−1^		kg.ha^−1^
TUTCEeS1	5.08 ± 0.12 bcd	3.29 ± 0.18 abc	167.61 ± 12.40 ab	−0.48 ± 0.17 e	132.84 ± 10.24 abc	0.60 ± 0.03 b	2.55 ± 0.26 abc	15.25 ± 1.44 bcd	0.67 ± 0.14 l	3.43 ± 0.56 ab	5.68 ± 0.14 bcd	5.84 ± 0.24 abc	182.86 ± 12.65 ab	−0.38 ± 0.16 fg	136.27 ± 10.03 abc
TUTCEeS2	5.78 ± 0.32 b	3.45 ± 0.14 ab	200.47 ± 18.53 a	−0.37 ± 0.30 e	165.70 ± 24.79 a	0.60 ± 0.06 b	2.42 ± 0.03 b–e	14.53 ± 1.55 b–e	2.05 ± 0.02 cd	2.71 ± 3.04 ab	6.38 ± 0.37 b	5.87 ± 0.17 abc	215.00 ± 20.01 a	−0.21 ± 0.28 efg	168.40 ± 27.83 a
TUTCEeS3	3.77 ± 0.38 fgh	2.91 ± 0.26 abc	111.40 ± 19.75 c–f	0.13 ± 0.02 de	76.62 ± 17.77 c–g	0.57 ± 0.10 b	2.52 ± 0.07 a–d	14.28 ± 2.48 b–e	1.64 ± 0.17 efg	2.45 ± 2.56 ab	4.33 ± 0.34 d–i	5.43 ± 0.31 bcd	125.67 ± 18.54 c–g	0.31 ± 0.03 d–g	79.08 ± 15.24 c–f
TUTCEeS4	4.05 ± 0.05 efg	2.63 ± 0.12 bcd	106.48 ± 5.83 def	0.58 ± 0.48 cde	71.70 ± 5.30 d–h	0.60 ± 0.03 b	2.12 ± 0.01 fg	12.72 ± 0.61 b–f	2.28 ± 0.06 bc	0.89 ± 2.30 ab	4.18 ± 0.49 e–i	4.75 ± 0.12 de	119.19 ± 5.23 d–g	0.76 ± 0.45 cde	72.60 ± 6.47 d–g
TUTCEeS5	4.82 ± 0.15 cde	3.55 ± 0.14 ab	170.49 ± 3.25 ab	0.02 ± 0.22 e	135.71 ± 4.82 abc	0.55 ± 0.05 b	2.37 ± 0.02 c–f	13.03 ± 1.24 b–f	1.30 ± 0.07 hij	1.21 ± 2.50 ab	5.37 ± 0.10 b–e	5.91 ± 0.13 abc	183.52 ± 3.23 ab	0.11 ± 0.20 efg	136.92 ± 6.80 abc
TUTCEeS6	4.87 ± 0.27 cde	3.93 ± 0.57 a	191.61 ± 31.60 a	0.13 ± 0.07 de	156.83 ± 23.94 ab	0.63 ± 0.07 b	2.76 ± 0.08 a	17.46 ± 1.95 b	1.19 ± 0.20 ijk	5.64 ± 0.96 a	5.50 ± 0.33 b–e	6.68 ± 0.64 a	209.07 ± 33.27 ab	0.22 ± 0.05 efg	162.48 ± 23.73 ab
TUTCEeS7	4.65 ± 0.14 c–f	3.70 ± 0.15 a	172.44 ± 12.41 ab	−0.57 ± 0.25 e	137.67 ± 17.01 ab	0.45 ± 0.03 b	2.52 ± 0.02 a–d	11.33 ± 0.67 c–f	0.73 ± 0.07 l	−0.49 ± 2.14 ab	5.10 ± 0.13 b–g	6.22 ± 0.17 ab	183.77 ± 12.03 ab	−0.49 ± 0.24 g	137.17 ± 18.54 abc
TUTCEeS8	4.57 ± 0.34 c–f	2.98 ± 0.02 abc	136.01 ± 10.89 b–e	−0.49 ± 0.11 e	101.24 ± 17.18 b–f	0.50 ± 0.03 b	2.52 ± 0.12 a–d	12.64 ± 1.08 b–f	1.21 ± 0.07 ijk	0.82 ± 1.85 ab	5.07 ± 0.37 b–g	5.50 ± 0.12 bcd	148.65 ± 11.71 b–e	−0.34 ± 0.10 efg	102.05 ± 19.02 b–e
TUTCEeS9	4.03 ± 0.24 efg	1.67 ± 0.26 de	66.78 ± 8.75 fgh	2.01 ± 0.71 ab	32.01 ± 4.04 ghi	0.57 ± 0.03 b	2.44 ± 0.10 b–e	13.86 ± 1.28 b–f	1.74 ± 0.17 ef	2.04 ± 3.09 ab	4.93 ± 0.58 c–g	4.11 ± 0.22 efg	80.65 ± 7.53 ghi	1.92 ± 0.59 ab	34.05 ± 4.72 fgh
TUTCEeS10	2.88 ± 0.07 i	2.93 ± 0.19 abc	84.27 ± 4.73 fgh	0.61 ± 0.06 cde	49.49 ± 4.46 f–i	0.58 ± 0.07 b	1.95 ± 0.01 ghi	11.37 ± 1.28 c–f	1.64 ± 0.04 efg	−0.45 ± 3.08 ab	3.43 ± 0.17 hij	4.88 ± 0.18 cde	95.64 ± 4.09 f–i	0.73 ± 0.08 c–f	49.04 ± 7.50 f–h
TUTCEeS11	2.63 ± 0.18 i	1.89 ± 0.45 de	51.33 ± 16.04 gh	1.78 ± 0.72 abc	16.56 ± 9.95 hi	0.58 ± 0.02 b	1.70 ± 0.09 hi	9.90 ± 0.68 ef	2.43 ± 0.03 b	−1.92 ± 1.25 ab	3.38 ± 0.36 hij	3.58 ± 0.53 g	61.23 ± 16.71 hi	1.86 ± 0.63 ab	14.64 ± 9.29 gh
TUTCEeS12	5.20 ± 0.20 bc	3.36 ± 0.04 ab	174.53 ± 5.13 ab	−0.42 ± 0.11 e	139.76 ± 3.36 ab	0.52 ± 0.02 b	2.23 ± 0.14 ef	11.50 ± 0.47 c–f	1.65 ± 0.20 efg	−0.32 ± 1.36 ab	5.75 ± 0.18 bc	5.59 ± 0.12 bcd	186.03 ± 5.59 ab	−0.29 ± 0.11 efg	139.44 ± 4.58 abc
TUTCEeS13	4.43 ± 0.14 c–f	3.41 ± 0.15 ab	151.53 ± 10.02 a–d	−0.13 ± 0.18 e	116.75 ± 14.99 a–e	0.50 ± 0.10 b	2.54 ± 0.02 abc	12.70 ± 2.66 b–f	1.08 ± 0.06 jk	0.88 ± 3.31 ab	4.93 ± 0.55 c–g	5.95 ± 0.16 ab	164.23 ± 8.10 a–d	−0.04 ± 0.17 efg	117.64 ± 15.97 a–d
TUTCEeS14	4.20 ± 0.15 def	3.81 ± 0.26 a	159.35 ± 5.69 abc	−0.41 ± 0.24 e	124.57 ± 11.32 a–d	0.42 ± 0.04 b	2.45 ± 0.01 b–e	10.18 ± 1.02 def	0.94 ± 0.01 kl	−1.64 ± 0.79 ab	4.62 ± 0.36 c–h	6.26 ± 0.27 ab	169.53 ± 5.27 abc	−0.33 ± 0.23 efg	122.94 ± 12.10 a–d
TUTCEeS15	4.92 ± 0.12 cde	3.14 ± 0.29 abc	154.81 ± 17.73 a–d	−0.15 ± 0.28 e	120.04 ± 24.24 a–d	0.60 ± 0.10 b	2.62 ± 0.05 abc	15.80 ± 2.96 bc	1.70 ± 0.02 efg	3.98 ± 2.39 ab	5.52 ± 0.04 b–e	5.76 ± 0.24 abc	170.61 ± 15.07 ab	0.03 ± 0.27 efg	124.02 ± 23.71 a–d
TUTCEeS16	4.55 ± 0.43 c–f	3.89 ± 0.56 a	174.27 ± 18.57 ab	−0.16 ± 0.24 e	139.49 ± 21.79 ab	0.50 ± 0.07 b	2.51 ± 0.12 a–d	12.60 ± 1.89 b–f	1.38 ± 0.16 g–j	0.78 ± 1.38 ab	5.05 ± 0.47 b–g	6.40 ± 0.45 ab	186.86 ± 19.93 abc	−0.06 ± 0.23 efg	140.27 ± 23.17 abc
TUTCEeS17	3.29 ± 0.43 ghi	2.30 ± 0.41 cde	76.67 ± 20.04 fgh	0.63 ± 0.60 cde	41.90 ± 27.79 ghi	0.50 ± 0.03 b	1.95 ± 0.04 ghi	9.75 ± 0.44 ef	1.81 ± 0.09 de	−2.07 ± 1.44 ab	3.79 ± 0.41 g–j	4.26 ± 0.37 efg	86.43 ± 19.92 ghi	0.75 ± 0.55 c–f	39.84 ± 28.48 fgh
TUTCEeS18	4.85 ± 0.13 cde	3.46 ± 0.40 ab	168.47 ± 22.48 ab	−0.22 ± 0.17 e	131.16 ± 30.74 abc	0.61 ± 0.04 b	2.66 ± 0.05 ab	16.26 ± 1.32 bc	2.17 ± 0.11 bc	4.44 ± 2.75 ab	5.46 ± 0.56 b–e	6.12 ± 0.45 ab	184.73 ± 23.77 ab	−0.01 ± 0.15 efg	135.60 ± 32.15 abc
TUTCEeS19	3.02 ± 0.34 hi	1.58 ± 0.07 e	47.91 ± 6.74 gh	1.30 ± 0.33 bcd	13.14 ± 9.16 hi	0.60 ± 0.03 b	1.68 ± 0.02 i	10.10 ± 0.57 def	2.19 ± 0.07 bc	−1.72 ± 2.20 ab	3.95 ± 0.70 f–j	3.26 ± 0.08 g	58.01 ± 7.21 hi	1.45 ± 0.28 bc	11.42 ± 11.25 gh
TUTCEeS21	3.13 ± 0.09 hi	1.63 ± 0.03 e	51.13 ± 0.79 gh	2.83 ± 0.03 a	16.36 ± 8.48 hi	0.53 ± 0.04 b	1.81 ± 0.07 hi	9.64 ± 0.67 ef	2.95 ± 0.07 a	−2.18 ± 1.91 b	3.33 ± 0.47 hij	3.45 ± 0.09 g	60.77 ± 1.43 hi	2.85 ± 0.03 a	14.18 ± 10.39 gh
TUTCEeS36	2.53 ± 0.04 i	1.73 ± 0.06 de	43.62 ± 0.78 i	2.67 ± 0.13 a	9.64 ± 7.39 i	0.48 ± 0.03 b	1.96 ± 0.05 gh	9.52 ± 0.88 ef	2.98 ± 0.01 a	−2.30 ± 2.48 b	2.80 ± 0.24 j	3.69 ± 0.06 fg	53.15 ± 0.87 i	2.72 ± 0.10 a	7.35 ± 9.86 h
TUTCEeS38	3.23 ± 0.32 ghi	2.89 ± 0.71 abc	94.97 ± 29.15 efg	0.42 ± 1.04 de	60.20 ± 34.47 e–g	0.52 ± 0.03 b	1.73 ± 0.07 hi	9.01 ± 0.92 f	1.45 ± 0.06 f–i	−2.82 ± 2.69 b	3.75 ± 0.35 g–j	4.62 ± 0.68 def	103.98 ± 29.21 e–h	0.46 ± 0.94 c–g	57.39 ± 35.79 e–h
B. XS21	4.64 ± 0.08 c–f	3.64 ± 0.45 ab	168.76 ± 19.79 ab	−0.47 ± 0.25 e	133.99 ± 22.27 abc	0.58 ± 0.02 b	2.25 ± 0.01 def	13.13 ± 0.44 b–f	1.57 ± 0.11 efg	1.31 ± 1.77 ab	5.23 ± 0.09 b–f	5.89 ± 0.44 abc	181.89 ± 19.36 ab	−0.31 ± 0.22 efg	135.29 ± 23.13 abc
Uninoculated (Control)	2.48 ± 0.60 i	1.40 ± 0.04 e	34.77 ± 8.03 i	1.25 ± 0.03 bcd	ND	0.62 ± 0.10 b	1.93 ± 0.07 ghi	11.82 ± 1.81 c–f	1.45 ± 0.04 f–i	ND	3.10 ± 0.70 ij	3.33 ± 0.05 g	46.59 ± 9.80 i	1.30 ± 0.02 bcd	ND
5 mM KNO_3_	7.70 ± 040 a	1.43 ± 0.13 e	110.26 ± 12.02 def	1.70 ± 0.04 abc	ND	1.42 ± 0.19 a	1.90 ± 0.05 ghi	26.78 ± 2.85 a	0.20 ± 0.02 m	ND	9.12 ± 0.58 a	3.33 ± 0.15 g	137.04 ± 13.17 b–f	1.41 ± 0.05 bc	ND
F statistics	19.78 ***	7.82 ***	12.31 ***	7.77 ***	8.16 ***	7.40 ***	16.36 ***	5.78 ***	42.43 ***	2.50 **	10.34 ***	13.01 ***	12.60 ***	8.75 ***	7.71 ***

**Table 5 microorganisms-11-02786-t005:** Carbon assimilation and δ^13^C of 22 jack bean isolates sourced from various locations in Eswatini; the plants were harvested 60 days after planting. Values (Mean ± SE) with dissimilar letters in a column are significantly different at *p* ≤ 0.05, ** *p* ≤ 0.01, *** *p* ≤ 0.001.

Isolate	Shoot				Root			
	C	C Content	C:N Ratio	δ^13^C	C	C Content	C:N Ratio	δ^13^C
	%	mg.plant^−1^		‰	%	mg.plant^−1^		‰
TUTCEeS1	43.18 ± 0.0.16 abc	2195.35 ± 58.07 bcd	13.20 ± 0.67 e	−23.98 ± 0.06 ab	44.73 ± 0.73 ab	268.76 ± 16.91 b	−23.48 ± 0.08 bcd	17.90 ± 1.90 fgh
TUTCEeS2	43.24 ± 0.11 abc	2500.50 ± 134.15 b	12.57 ± 0.55 e	−24.59 ± 0.13 b–f	42.70 ± 0.69 a–f	255.52 ± 21.18 b	−24.01 ± 0.27 e–h	17.68 ± 0.51 fgh
TUTCEeS3	42.89 ± 0.07 abc	1615.54 ± 160.71 fgh	15.00 ± 1.47 de	−24.24 ± 0.18 a–d	43.72 ± 0.94 abc	249.49 ± 49.03 b	−24.13 ± 0.24 fgh	17.36 ± 0.49 fgh
TUTCEeS4	42.59 ± 0.39 c	1725.10 ± 35.70 efg	16.28 ± 0.73 de	−25.35 ± 0.20 g–j	43.74 ± 0.18 abc	262.52 ± 13.44 b	−23.62 ± 0.07 c–f	20.64 ± 0.13 de
TUTCEeS5	42.76 ± 0.11 bc	2059.18 ± 57.21 cde	12.09 ± 0.46 e	−24.31 ± 0.30 a–d	42.92 ± 0.27 a–f	236.08 ± 21.65 b	−23.10 ± 0.07 ab	18.13 ± 0.23 fg
TUTCEeS6	43.29 ± 0.14 abc	2107.25 ± 119.07 cde	11.44 ± 1.43 e	−24.23 ± 0.19 a–d	43.43 ± 0.32 a–d	275.25 ± 29.81 b	−23.11 ± 0.20 ab	15.78 ± 0.33 h
TUTCEeS7	42.44 ± 0.17 cd	1973.29 ± 60.44 c–f	11.51 ± 0.48 e	−24.60 ± 0.08 b–f	42.58 ± 0.34 a–f	191.80 ± 13.70 b	−23.31 ± 0.07 bc	16.91 ± 0.27 fgh
TUTCEeS8	43.20 ± 0.12 abc	1972.79 ± 145.71 c–f	14.52 ± 0.09 de	−23.87 ± 0.30 a	45.06 ± 0.03 a	225.33 ± 13.16 b	−23.56 ± 0.17 b–e	17.95 ± 0.85 fgh
TUTCEeS9	40.82 ± 0.28 ef	1646.78 ± 99.89 fg	25.49 ± 3.59 abc	−25.63 ± 0.16 h–k	42.03 ± 1.99 b–f	237.40 ± 12.74 b	−23.05 ± 0.12 ab	17.30 ± 1.08 fgh
TUTCEeS10	43.14 ± 0.06 abc	1243.84 ± 32.35 i	14.86 ± 0.96 de	−24.56 ± 0.23 b–f	40.97 ± 0.67 c–g	238.21 ± 24.48 b	−24.97 ± 0.09 i	21.01 ± 0.27 d
TUTCEeS11	40.99 ± 0.85 ef	1082.07 ± 95.94 i	23.73 ± 4.24 bc	−25.70 ± 0.19 ijk	42.01 ± 0.42 b–f	244.99 ± 6.61 b	−24.29 ± 0.11 h	24.88 ± 1.07 a
TUTCEeS12	43.88 ± 0.03 a	2282.06 ± 89.20 bc	13.07 ± 0.16 e	−24.47 ± 0.13 a–e	39.02 ± 2.38 g	200.81 ± 6.31 b	−23.51 ± 0.09 b–e	17.49 ± 0.50 fgh
TUTCEeS13	42.943 ± 0.19 abc	1903.96 ± 61.73 c–f	12.63 ± 0.50 e	−24.15 ± 0.15 a–d	40.70 ± 1.32 d–g	201.33 ± 37.63 b	−23.72 ± 0.12 c–g	16.03 ± 0.61 gh
TUTCEeS14	43.68 ± 0.32 ab	1833.66 ± 53.01 def	11.56 ± 0.72 e	−24.50 ± 0.09 b–e	43.29 ± 0.26 a–d	180.28 ± 18.72 b	−23.73 ± 0.11 c–g	17.69 ± 0.09 fgh
TUTCEeS15	42.80 ± 0.07 bc	2104.23 ± 48.15 cde	13.92 ± 1.42 de	−24.08 ± 0.17 abc	43.62 ± 0.74 a–d	263.24 ± 49.37 b	−23.99 ± 0.16 d–h	16.65 ± 0.18 fgh
TUTCEeS16	43.35 ± 0.26 abc	1971.24 ± 181.74 c–f	11.54 ± 1.41 e	−24.78 ± 0.23 d–g	43.19 ± 0.16 a–e	217.62 ± 32.30 b	−23.46 ± 0.26 bc	17.29 ± 0.85 fgh
TUTCEeS17	41.54 ± 0.46 de	1368.99 ± 184.52 ghi	19.77 ± 3.29 cd	−25.34 ± 0.20 g–j	41.62 ± 0.49 c–g	207.86 ± 10.24 b	−23.66 ± 0.14 c–g	21.30 ± 0.188 cd
TUTCEeS18	42.72 ± 0.04 bc	2071.89 ± 55.57 cde	12.64 ± 1.31 e	−24.52 ± 0.17 b–e	42.77 ± 0.49 a–f	261.24 ± 19.22 b	−22.73 ± 0.11 a	16.08 ± 0.12 gh
TUTCEeS19	40.51 ± 0.36 f	1224.05 ± 149.12 i	25.74 ± 1.20 abc	−25.61 ± 0.03 h–k	40.24 ± 0.58 efg	241.73 ± 14.91 b	−23.79 ± 0.24 c–g	23.92 ± 0.34 a
TUTCEeS21	40.76 ± 0.41 ef	1277.70 ± 49.46 hi	24.97 ± 0.68 abc	−25.85 ± 0.21 kj	42.17 ± 0.08 a–f	224.87 ± 18.27 b	−23.44 ± 0.08 bc	23.33 ± 0.98 abc
TUTCEeS36	41.55 ± 0.08 de	1049.74 ± 18.17 i	24.09 ± 0.80 bc	−25.51 ± 0.30 h–k	40.06 ± 1.05 fg	193.51 ± 13.35 b	−23.69 ± 0.16 c–g	20.41 ± 0.66 de
TUTCEeS38	42.37 ± 0.57 cd	1372.30 ± 148.39 ghi	17.08 ± 5.00 de	−25.16 ± 0.28 f–i	41.15 ± 0.85 c–g	213.16 ± 17.81 b	−23.70 ± 0.04 c–g	23.77 ± 0.54 ab
B. XS21	42.75 ± 0.25 bc	1984.85 ± 34.03 c–f	12.06 ± 1.28 e	−25.02 ± 0.02 e–h	42.38 ± 0.86 a–f	247.21 ± 9.03 b	−24.17 ± 0.22 gh	18.83 ± 0.35 ef
Uninoculated (Control)	40.22 ± 0.19 f	998.97 ± 242.14 i	28.70 ± 0.94 abc	−26.08 ± 0.17 k	41.96 ± 0.45 b–f	258.20 ± 41.18 b	−24.86 ± 0.17 i	21.79 ± 0.58 bcd
5 mM KNO_3_	42.85 ± 0.15 bc	3300.42 ± 179.91 a	30.42 ± 2.56 a	−24.62 ± 0.17 c–f	40.84 ± 0.45 c–g	577.50 ± 73.28 a	−23.07 ± 0.11 ab	21.48 ± 0.46 cd
F statistics	12.20 ***	20.57 ***	10.37 ***	11.50 ***	2.9 **	6.88 ***	11.90 ***	16.64 ***

## Data Availability

Data are contained within the article.

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
