# Peer review of "Symbiotic Functioning and Photosynthetic Rates Induced by Rhizobia Associated with Jack Bean (Canavalia ensiformis L.) Nodulation in Eswatini"

_microorganisms, 2023, doi:10.3390/microorganisms11112786_

Round 1
Reviewer 1 Report
Comments and Suggestions for Authors
The manuscript entitled "Symbiotic Functioning and Photosynthetic Rates Induced by Rhizobia Associated with Jack Bean (Canavalia ensiformis L.) Nodulation in Eswatini" is interesting and important for African agriculture.
However, there are some points which should be revised as follows.
The %RSE is not a good indicator of the percentage dependence of N2 fixation, because some data exhibited over 100%. Also, the authors used sterile sand culture with non-inoculated control plants, so the difference in N contents between this control and the inoculated plants is a reliable indicator for N2 fixation. I recommend deleting the data of %RSE.
If you would like to use an indicator of %RSA, please show the correlation between the difference in N contents between the non-inoculated control and inoculated plants and %RSA.
Line 68; The literatures [15,28,29,30,31,32,33] should be changed to [15,28-33].
Table 1; Please show the method for available P analysis.
Line 206; Table 3 is Table 2.
Line 255; %RSA (124.64%) is not possible. The highest value should be 100%.
Figure 3; In the figures of the correlation analysis the horizontal axis is usually the cause of the vertical axis. Please change as follows;
A; horizontal axis shoot dry matter, vertical axis photosynthetic rate.
B; horizontal axis N content, vertical axis shoot dry matter.
C; horizontal axis fixed N, vertical axis shoot dry matter.
Figure 5; The same as in Figure 3.
Comments on the Quality of English Language
The quality of English looks good.
Author Response
Comments and Suggestions for Authors
The manuscript entitled "Symbiotic Functioning and Photosynthetic Rates Induced by Rhizobia Associated with Jack Bean (Canavalia ensiformis L.) Nodulation in Eswatini" is interesting and important for African agriculture.
However, there are some points which should be revised as follows.
- REVIEWER #1: The %RSE is not a good indicator of the percentage dependence of N2 fixation, because some data exhibited over 100%. Also, the authors used sterile sand culture with non-inoculated control plants, so the difference in N contents between this control and the inoculated plants is a reliable indicator for N2 fixation. I recommend deleting the data of %RSE.
- AUTHORS RESPONSE:
We differ with Reviewer #1 on the comments that “The %RSE is not a good indicator of the percentage dependence of N2 fixation, because some data exhibited over 100%”. This is the standard equation for calculating %RSE in the literature, and %RSE values can be above 100% [42-44].
- REVIEWER #1: If you would like to use an indicator of %RSA, please show the correlation between the difference in N contents between the non-inoculated control and inoculated plants and %RSE.
- AUTHORS RESPONSE: The correlation for the difference in N contents between the non-inoculated control and inoculated plants vs. %RSE has been done as suggested above by Reviewer #1, and it is significant (see text). The %RSE data are therefore retained.
- REVIEWER #1: Line 68; The literatures [15,28,29,30,31,32,33] should be changed to [15,28-33].
- AUTHORS RESPONSE: It’s done (see text).
- REVIEWER #1: Table 1; Please show the method for available P analysis.
- AUTHORS RESPONSE: It’s done (see text). “The chemical properties of the soils used in this study, including available P, were measured as described by [37] presented in Table 1”.
- REVIEWER #1: Line 206; Table 3 is Table 2.
- AUTHORS RESPONSE: It’s done (see text).
- REVIEWER #1: Line 255; %RSE (124.64%) is not possible. The highest value should be 100%.
- AUTHORS RESPONSE: It is possible (see authors response above).
- REVIEWER #1: Figure 3; In the figures of the correlation analysis the horizontal axis is usually the cause of the vertical axis. Please change as follows;
A; horizontal axis shoot dry matter, vertical axis photosynthetic rate.
B; horizontal axis N content, vertical axis shoot dry matter.
C; horizontal axis fixed N, vertical axis shoot dry matter.
- Figure 5; The same as in Figure 3.
- AUTHORS RESPONSE: All done, as requested by Reviewer #1 (see text).

Reviewer 2 Report
Comments and Suggestions for Authors
The authors tried to identify effective rhizobial strains for increased jack bean production.
However, I suppose the authors should present the data for their own purpose. The manuscript seems like there are two goals.
The abstract and discussion sections must be rewritten entirely.
My comments are in the file.
Thanks,

Author Response
Reviewer 2
- REVIEWER #2:
Comments and Suggestions for Authors
The authors tried to identify effective rhizobial strains for increased jack bean production.
However, I suppose the authors should present the data for their own purpose. The manuscript seems like there are two goals.
The abstract and discussion sections must be rewritten entirely.
My comments are in the file.
Thanks
- REVIEWER #2:
The above guidance/critique provided by Reviewer #2 is not clear enough for the authors to improve the manuscript. Comments like the “The abstract and discussion sections must be rewritten entirely” has not pointed out the direction in which to re-write the Abstract and Discussion.
- Response to Comments by Reviewer #2 in Text:
Abstract:
Comment 1: The title doesn’t tell all the contents
Authors response: Symbiotic functioning covers all aspects of N2 fixation mentioned in the Abstract, and its relationship with photosynthesis.
Comment 2: This M & M Section needs to be concise.
Authors response: The sentence starting “To assess the N2-fixing efficiency” … up to the ending “was used for 15N/14N analysis” has been deleted for brevity and conciseness.
Comment 3: This M & M Section needs to be concise.
Authors response: See response above. See response above. The last sentence of the Abstract has however been retained as it contains vital information about isolate N2-fixing efficiency.
Introduction:
Comment 4/Response 4: “g/kg” has been corrected to “g.kg-1“.
Comment 5/Response 5: “5mL” has been changed to “5 mL”, and “4 oC” to “4oC”.
Comment 6/Response 6: Under Figure 1: “strain” has been change to “strain.”
Comment 7: In legend of Table 2, “I suppose full name is better”.
Response 7: It’s done.
Comment 8: In Table 2, “g/plant” is not SI unit.
Response 8: In Table 2, we have “g.plant-1” (which is an SI unit) and not “g/plant”.
Comment 9: In Table 3, for “A, gs, E, WUE”, full name is better”.
Response 9: It’s done.
Comment 10: In Table 3, change “umol (CO2)” to “mol (CO2)”.
Response 10: It’s done.
Comment 11: In Table 3, change “1,20” to “1.20”
Response 11: It’s done for ALL tables in the paper.
Comment 12: In Table 4, for whole-plant, “Is this data necessary?”
Response 12: Yes, it is necessary.
Comment 13: For numerical data of parameters in Table 4, “You must have comma”
Response 13: It’s done for ALL tables in the paper.
Comment 14: In Table 5, “mg plant-1” for C content.
Response 14: In Table 5, “mg plant-1” for C content has been changed to “mg.plant-1”, with a decimal in Table 5 and for ALL other tables.
Comment 15: Move section 4.1 on “Bioprospecting for superior rhizobia” to Introduction.
Response 15: The section under “Bioprospecting for superior rhizobia” has been moved to the Introduction, and is the fourth paragraph of the Introduction.
Comment 16: Reviewer #2 has recommended that Section 4.2 on Symbiotic effectiveness and its effect on photosynthesis be moved to Results. In fact, Reviewer #2 does not think that the data is needed as it is not our goal in this study.
Response 16: We disagree with Reviewer #2 on both points. So, we are retaining Section 4.2 of the Discussion as is.
Comment 17: Reviewer #2 has indicated that “There are not enough citations to support the findings of this support”.
Response 17: Indeed, Reviewer #2 is right! Jack bean is a neglected, under-utilized and under-researched food grain legume, with little information on it globally. Therefore, there are very few published papers to cite.

Round 2
Reviewer 1 Report
Comments and Suggestions for Authors
The manuscript has been well-revised following the reviewer's questions and comments.
Adding the equation of %RSE in lines 145-146 is good for understanding the meaning of the relative symbiotic effectiveness.
I misunderstood the meaning of relative symbiotic effectiveness, but now I know that %RSE is over 100%.
However, please revise the equation, because the underlines are not placed in the correct place.
Also please add RSE in line 141.
The relative symbiotic effectiveness (RSE) which is defined ...
Author Response
Reviewer 1
The manuscript has been well-revised following the reviewer's questions and comments.
Adding the equation of %RSE in lines 145-146 is good for understanding the meaning of the relative symbiotic effectiveness.
I misunderstood the meaning of relative symbiotic effectiveness, but now I know that %RSE is over 100%.
However, please revise the equation, because the underlines are not placed in the correct place.
- It’s done.
Also please add RSE in line 141.
The relative symbiotic effectiveness (RSE) which is defined ...
- It’s done.
Reviewer 2 Report
Comments and Suggestions for Authors
Thank you for the responses.
However, in the response 16, I don't agree with the authors on both points.
The authors should consider the meaning of 'discussion' and the goal of the study.
For instance, is correlation between shoot dry matter and photosynthetic rate the goal??
I don't wanna argue about this any further.
I would appreciate it if the editor decided.
Thanks,
Author Response
Reviewer 2
Thank you for the responses.
However, in the response 16, I don't agree with the authors on both points.
The authors should consider the meaning of 'discussion' and the goal of the study.
For instance, is correlation between shoot dry matter and photosynthetic rate the goal??
I don't wanna argue about this any further.
I would appreciate it if the editor decided.
Thanks,
- Thank you for the feedback.